# Tribological Behavior of Lamellar Molybdenum Trioxide as a Lubricant Additive

**DOI:** 10.3390/ma11122427

**Published:** 2018-11-30

**Authors:** Wei Tang, Rui Liu, Xiangyong Lu, Shaogang Zhang, Songyong Liu

**Affiliations:** School of Mechatronic Engineering, China University of Mining and Technology, Xuzhou 221116, Jiangsu, China; lr12343@163.com (R.L.); luxy1994@163.com (X.L.); chensicumt@126.com (S.Z.); liusongyong@163.com (S.L.)

**Keywords:** lamellar, MoO_3_, lubricant additives, friction-reducing, antiwear

## Abstract

In this study, the tribological behavior of lamellar MoO_3_ as a lubricant additive was investigated under different concentrations, particle sizes, normal loads, velocity, and temperature. The friction and wear tests were performed using a tribometer and with a reciprocating motion. The results indicate that the friction-reducing ability and antiwear property of the base oil can be improved effectively with the addition of lamellar MoO_3_. The 0.5 wt % and 0.1 wt % concentrations of MoO_3_ yield the best antifriction and antiwear effects, respectively. The maximum friction and wear reduction is 19.8% and 55.9%, compared with that of the base oil. It is also found the MoO_3_ additive can decrease the friction considerably under a high velocity and normal load, and increase the working temperature. The smaller the size of MoO_3_, the better the friction-reducing effect the lamellar MoO_3_ shows. The friction-reducing and antiwear mechanisms of lamellar MoO_3_ were discussed.

## 1. Introduction

Lubrication has been widely used in engineering applications to save energy and to improve the efficiency of the machines. Various additives have been studied to improve the properties of lubricants. The two-dimensional (2D) materials are commonly used as solid lubricants, such as graphene, MoS_2_ and Boron Nitride [1,2,3]. The excellent tribological performance of 2D nanoparticle materials are due to their lamellar structures [4,5,6,7]. The weak Van der Waals interlayer bonding leads to the adjacent layers slide easily against each other under the low shear strength. The strong covalent in-layer bonding makes the monolayer structure strong. 

Recently, studies have shown that 2D nanoparticles as additives can improve the friction-reducing and antiwear properties of base oil efficiency. He et al. [8] found that 2D nanoparticles of Y_2_O_3_ can improve lubrication and reduce viscosity efficiency as an effective lubricant additive. The 0.1 wt % of the Y_2_O_3_ was capable of reducing friction and viscosity as much as by 40% and 5%, respectively. From the investigation of Ingole [9], it was found that 0.25 wt % nano-TiO_2_ particles, as an additive to the base oil, slightly reduce the coefficient of friction from 0.1 to 0.09, stabilize the coefficient of friction around 0.09, and reduce the variability of friction curve. Battez et al. [10,11] indicated that 0.5% nano-ZnO, 0.5% nano-ZrO_2_, and 1% nano-CuO can reduce friction by about 21%, 22%, and 19%, respectively, and reduce the wear by about 55%, 55%, and 47%, respectively, when used as lubricant additives. The studies of Huang [12] showed that the tribological performance of magnetic fluids with proper Fe_3_O_4_ nanoparticles can be improved significantly. Fe_3_O_4_ nanoparticles of 4 wt % could reduce the wear scar diameter from 0.68 mm to 0.53 mm, and decrease the friction coefficient by about 31.3%. Some studies have shown that TiO_2_ nanoparticles and Al_2_O_3_ nanoparticles as additives have good antiwear and antifriction properties, respectively. The addition of TiO_2_ nanoparticles into water significantly reduces the friction coefficient, and improves the wear resistance. The friction coefficient and wear can be decreased by 49.5% and 97.8% [13]. Through the addition of a 10% volume of Al_2_O_3_ nanoparticles to an acetate buffer solution, it was possible to reduce the coefficient of friction by a factor of 2, and the stainless steel wear by a factor of 10 [14]. The addition of a 2D nanoparticles lubricant additive is also beneficial for environmental protection, because it might reduce the use of phosphorus-containing additives. It is also significantly less toxicity than organic additives [9,15,16]. These studies have proven that the 2D nanoparticles lubricant additive have become promising new lubricant materials. 

MoO_3_ has the same 2D layered structure as that of the above-mentioned laminar oxide and MoS_2_. It is hypothesized that lamellar MoO_3_ is able to reduce friction and wear as lubricant additives. Layered MoO_3_ has already shown significant applications in various fields of optics, electrochemistry, electronics, and sensors [17,18,19]. However, in previous reports, lamellar MoO_3_ was rarely studied as a lubricant additive.

Therefore, the present work describes the detailed study of the tribological behavior of lamellar MoO_3_ as a lubricant additive under different concentrations, particle sizes, normal loads, velocities, and temperatures. Friction and wear tests were carried out using a tribometer. Oleic acid was used as a surfactant to improve the dispersing properties of the lamellar MoO_3_. The coefficient of friction, wear scar depth and width, and elemental analysis of the tribofilms were discussed under different lubricated conditions. In addition, the related lubricant mechanism was discussed. The objective of this study is to understand the friction-reducing and antiwear effects and the related lubrication mechanisms of the MoO_3_. It is expected that this research can confirm an alternative 2D additive material in lubrication. 

## 2. Experimental Details

### 2.1. Friction and Wear Tests 

The friction and wear tests were carried out using a tribometer (UMT-2, Center for Tribology Inc., Campbell, CA, USA) with a pin-on-disk and a ball-on-disk configuration. The schematic of the testing system is shown in Figure 1. The reciprocating friction and wear tests were carried out under different lubricated conditions. The average test values were obtained based on the four-time repeated test.

Before testing, all friction pairs were cleaned with anhydrous ethanol under ultrasonication. During the test, a 0.2 mL oil sample was added to the sample surface every 30 min to ensure good lubrication. After the wear test, the samples were cleaned and dried to measure the width and depth of the wear scar, using a surface roughness tester (Taimig, Shanghai, China). The final width and depth were the averages of five points on the wear scar.

Table 1 shows the parameters of the pin-on-disk and ball-on-disk configurations. Table 2 shows the material chemical compositions of the pin-on-disk and ball-on-disk configurations.

### 2.2. Characterization

A scanning electron microscope (SEM, FEI, Hillsboro, AL, USA) was used to image the MoO_3_ samples. The morphology of the wear scars and their 2D profiles were analyzed with a high-powered microscope (YSDS, Beijing, China) and a surface roughness tester. The tip of the surface profile meter (Taimig, Shanghai, China) scanned across the surface along the sampling length and the surface profile was obtained. Elemental analysis of the tribofilms formed on the sample was performed using energy dispersive X-ray spectroscopy (EDS, FEI, Hillsboro, AL, USA).

### 2.3. Lubrication Preparation

Hydrothermally synthesized lamellar MoO_3_ (>99.90%) was used in this research (Fortuneibo-tech Co., Ltd., Shanghai, China). Figure 2 shows the images of MoO_3_. MoO_3_ was mixed with a base oil (liquid paraffin supplied by Sigma-Aldrich, (Shanghai, China) to obtain the different lubricant concentrations. According to the literature on the tribological behaviors of layered nanoparticles as lubricant additives [8,9,10,11,20], most concentrations that showed good antifriction and antiwear were around 0.1–0.5 wt %. So, the four final concentrations of MoO_3_ in base oil (0.1 wt %, 0.5 wt %, 1.0 wt %, and 1.5 wt %) were chosen. 

MoO_3_ was distributed into the base oil by mechanical stirring for 30 min. Oleic acid was then gradually added into the solution, followed by high-speed centrifugation at 15,000 rpm for 30 min to prepare the dispersive solutions. Oleic acid works as a surfactant of MoO_3_ in order to improve the dispersing properties of the lamellar MoO_3_ [21]. The suspension was then processed by ultrasonication with stirring for 30 min to breakdown any remaining agglomeration, leading to the desired samples with different contents of MoO_3_. 

## 3. Results and Discussion

### 3.1. Influence of the Concentration of MoO_3_

The friction and wear tests were carried out with different concentrations of MoO_3_ (0 wt %, 0.1 wt %, 0.5 wt %, 1.0 wt %, and 1.5 wt %) under a velocity of 80 mm/s, a normal load of 200 N, a sliding distance of 10 mm, and normal ambient conditions (25 °C, relative humidity (RH) 50–60%). The test conditions were chosen according to the references [22,23]. The friction and wear test was carried out for 0.5 hr and 8 hr, respectively. The size of MoO_3_ is approximately 2~10 μm.

Figure 3 shows the typical friction coefficients of the base oil and base oil with different concentrations of MoO_3_. The typical friction coefficient curves suggest that MoO_3_ can reduce variability and stabilize friction. Table 3 shows the average friction coefficients of different concentrations of MoO_3_ based on four times tests. Table 4 is the variance analysis for different concentrations of MoO_3_. Concentrations of 0 wt %, 0.1 wt %, 0.5 wt %, and 1 wt % MoO_3_ were significantly different from each other (*p* < 0.05). The results show that with the additions of 0.1 wt %, 0.5 wt %, 1 wt %, and 1.5 wt % MoO_3_, the friction coefficient was reduced by approximately 6%, 15%, 9%, and 7%, compared with that of base oil. These results indicate that MoO_3_ can effectively reduce the friction coefficient of the lubricant, and the 0.5 wt % concentration of MoO_3_ yields the best antifriction performance. The higher the concentration is, the more the microsheet particles enter into the contact area, resulting in better lubrication. However, when the concentration of MoO_3_ is too high, the high surface energy makes the particles agglomerate together and form clusters. The clusters prevent the particles from entering the contact area, and they decrease the lubrication effect. 

Figure 4 shows the width and depth of the wear scar of different concentrations of MoO_3_. The base oil showed the largest wear scar, with a width of 671.29 µm and a depth of 9.28 µm. The oil with 0.5 wt % MoO_3_ shows the smallest wear scar with a width of 575.67 µm, which was decreased by 14.2% compared with that of the base oil. The oil with 0.1 wt % MoO_3_ showed the smallest wear scar depth at 4.09 µm, which was decreased by 55.9%, compared with that of the base oil. The results suggest that, as a lubricant additive, MoO_3_ showed good antiwear properties. The 0.1 wt % MoO_3_ shows the optimal antiwear effect.

The surface images and section profiles of the wear scar are shown in Figure 5. From the section profiles of the wear scar, the heights of the bulge were 9.1 µm, 3.8 µm, 1.8 µm, 3.3 µm, and 2.3 µm, respectively. The depths of the cavity were 8.9 µm, 3.7 µm, 5.5 µm, 5.8 µm, and 6.1 µm, respectively. From the surface images, the surface lubricated by the base oil also showed more and deeper furrows than the surface lubricated by the oil with MoO_3_ additive. The results indicated that there was serious abrasive wear, and plastic deformation occurred on the contact surface. In comparison, the surfaces lubricated by oil with MoO_3_ additive showed mild wear and less plastic deformation.

The EDS data of the wear scars lubricated by the base oil and different concentrations of MoO_3_ additives are shown in Figure 6. The results show that all surfaces lubricated by oil with MoO_3_ additive have a proportion of Mo elements, indicating that the tribofilm is formed on the contact surface. 

The good friction-reducing and antiwear performance of MoO_3_ can be explained as follows. First, the microscopic size and lamellar shape allow MoO_3_ to easily enter into the contact area of the sliding surface, as shown in Figure 7. During the relative motion, a shear stress is applied to the contact material and lubricant. The layered structure of MoO_3_ can easily produce interlayer shearing, which forms a sliding system in the contact area. Compared with the hard contact, the sliding contact has much lower friction. Second, it is known that the surface of material is uneven and very rough on the micro- and nanoscales. The micro-layered MoO_3_ can fill the concave areas and gaps between the contact surfaces, and thus smooth the surfaces. The smooth surface can reduce the contact pressure and plastic deformation, which improves the lubrication effect. 

Third, the physical film and tribofilm could form at the contact area. The schematic diagrams of the physical film and tribofilm formations are shown in Figure 8. First, due to the high surface energy and easy shearing, MoO_3_ is adsorbed on the contact surface and it forms a physical protective film, which prevents the direct contact of the two sliding surfaces, and it reduces friction and wear. However, during the sliding process, the physical film is easily fractured. The heat generated during friction promotes the tribochemical reaction between the MoO_3_, the base oil, and the metal. The dense and stable oxidizing tribofilm replaces the physical film, which improves tribological performance considerably.

In addition, the layered structure of micro-MoO_3_ can help the base oil to form a continuous oil film. All these reasons lead to the excellent lubrication of MoO_3_. 

### 3.2. Influence of the Normal Load and Velocity 

The influence of the normal load was tested with 0.5 wt % MoO_3_ under a velocity of 80 mm/s, normal ambient conditions (25 °C, relative humidity (RH) 50–60%), and different normal loads (50 N, 100 N, 150 N, 200 N, and 250 N). The variation range of the normal load was chosen according to the references [23,24,25].

The influence of velocity was tested with 0.5 wt % MoO_3_ under a normal load of 200 N, normal ambient conditions (25 °C, RH 50–60%), and different velocities (20 mm/s, 40 mm/s, 60 mm/s, 80 mm/s, and 100 mm/s). The variation range of the velocity was chosen according to the references [25,26]. The size of MoO_3_ is approximately 2~10 μm.

Figure 9 shows the friction coefficients of the base oil with 0.5 wt % MoO_3_ under different normal loads. It shows that the friction coefficient is reduced by approximately 5.1%, 6.9%, 12.4%, 15.7%, and 19.8% under applied loads of 50 N, 100 N, 150 N, 200 N, and 250 N, respectively, compared with that of base oil. The friction coefficient of oil with 0.5 wt % MoO_3_ additives decreased slightly with the increase of the normal load. The friction coefficient of the base oil increased slightly with the increase of the normal load. 

Figure 10 shows the friction coefficients of base oil with 0.5 wt % MoO_3_ under different velocities. The results show that the friction coefficients of the two oil samples both decreased with the increase of velocity. The friction coefficient was decreased by 11.4%, 15.8%, 13.9%, 12.7%, and 7.8% at the velocities of 20 mm/s, 40 mm/s, 60 mm/s, 80 mm/s, and 100 mm/s compared with that of the base oil. The addition of MoO_3_ can reduce the friction coefficient in the case of variable velocity or load, indicating its good friction-reducing performance. 

The ratio of film thickness to surface roughness *λ* is usually used to determine the types of lubrication [27,28,29,30]. The formula of *λ* is expressed as follows:(1)$$\lambda =\frac{h_{min}}{\sqrt{R_{a1}{}^{2}+R_{a2}{}^{2}}}$$
where *R*_*a*1_ and *R*_*a*2_ are the roughness parameters (arithmetical mean heights that indicate the average of the absolute value along the sampling length), and *h_min_* is the minimum thickness of the oil film. Least squares were used to estimate the parameters of a regression equation. The regression equation of *h_min_* is as follows: (2)$$H_{min}=2.8U^{0.65}W^{-0.21}\;H_{min}=h_{min}/R$$
where *U* is the velocity influence factor and *W* is the load factor; *U* and *W* are expressed as follows:(3)$$U=\frac{\eta u}{E^{*}R}$$
(4)$$W=\frac{F}{E^{*}R^{2}}$$
(5)$$\frac{1}{E^{*}}=\frac{1-\upsilon_{1}{}^{2}}{E_{1}}+\frac{1-\upsilon_{2}{}^{2}}{E_{2}}$$
where *R* is the radius of the pin, *η* is the dynamic viscosity of the oil, *u* is the relative velocity of the two friction pairs, *F* is the load, *E** is the equivalent elastic modulus, *ν*_1_ and *ν*_2_ are the Poisson’s ratio of the two contact surfaces, and *E***_1_** and *E*_2_ are the elastic modulus of the two contact surfaces. Then, the expression of the minimum thickness of the oil film, *h*_min_, can be simplified as:(6)$$h_{min}=2.8{\left(\eta u\right)}^{0.65}\cdot F^{-0.21}\cdot E^{{*}^{-0.44}}\cdot R^{0.77}$$

According to Equation (5), the minimum thickness of the oil film was related with viscosity, load, and velocity. With the increase of the normal load, the thickness of the oil film decreases, which results in the increase of the friction coefficient. Micro-lamellar MoO_3_ smoothens the contact surfaces and serves as lubrication under the high normal load, and reduces friction and wear. With the increase of velocity, the thickness of the minimum oil film increases. Thus, more of the lubricant enters into the contact area of the sliding surface to improve the lubrication conditions. Moreover, due to the shear thinning characteristics, the layered lubricant additive can reduce the viscosity of the oil significantly with the increase of shear rate [8], which can also help to reduce the inner friction of the oil film and improve lubrication. 

### 3.3. Stribeck Curves

To plot Stribeck curves, velocity was increased from 10 mm/s to 600 mm/s under applied loads of 5 N with reciprocating motion [27,28,29,30]. 

Figure 11 shows the Stribeck curves of base oil with the addition of 0.5 wt % MoO_3_. The three lubrication regions (Region I: boundary lubrication, region II: mixed lubrication, and region III: hydrodynamic lubrication) determined by the coefficient of friction are labeled. The results suggest that the MoO_3_ additives can increase the bottom width of the mixed lubrication region and decrease the coefficient of friction in all lubrication regions, indicating its excellent friction-reducing performance and good load-carrying capacity. 

### 3.4. Influence of the Temperature 

The test was carried out with 0.5 wt % MoO_3_ under a velocity of 80 mm/s, a normal load of 200 N, and different temperatures (25 °C, 50 °C, 100 °C, 150 °C, and 200 °C). The variation range of the temperature was chosen according to the reference [7]. The temperature is controlled and monitored by the thermoelement of tribometer located in a vessel as shown in Figure 1. The size of MoO_3_ is approximately 2~10 μm.

Figure 12 shows the friction coefficients of the base oil and base oil with 0.5 wt % MoO_3_ under different temperatures. The results show that the critical temperature was 100°C and 150°C for the base oil and base oil with 0.5 wt % micro-MoO_3_ additives, respectively, indicating that the MoO_3_ additives can increase the working temperature of the oil. Below the critical temperature, the friction coefficient of oil with 0.5 wt % MoO_3_ additives was decreased by 12.7%, 15%, and 12.8% at the temperatures of 25 °C, 50 °C, and 100 °C, respectively, compared with that of the base oil, indicating the good friction-reducing performance of MoO_3_. Above the critical temperature, the friction coefficient increases sharply and clearly fluctuates, as shown in Figure 13.

Figure 14 shows the wear scar images and 2D profile curves of samples lubricated with the base oil and oil with 0.5 wt % MoO_3_ at 150 °C. In Figure 14a, it can be seen that there are deep furrows on the sample surfaces at 150 °C, indicating that the oil has lost its lubrication effect. Due to the viscosity reduction and oil evaporation with the increase of temperature, the intermolecular forces in the oil are weakened, and the lubricating film cannot be formed during sliding; thus, friction and wear increase greatly. However, at high temperature, the layered MoO_3_ additives still can be attached to the contact surfaces and function as lubrication. 

### 3.5. Influence of the Particle Sizes 

The test was carried out with 0.5 wt % MoO_3_ under a velocity of 80 mm/s, a normal load of 200 N, normal ambient conditions (25 °C, relative humidity (RH) 50–60%), and with different sizes and shapes of MoO_3_ (2~10 μm lamellar, 80~100 μm lamellar, and 100~300 μm block).

Figure 15 shows the SEM images of MoO_3_ particles with different sizes. Figure 16a shows the friction coefficients of differently sized MoO_3_. The results indicate that the friction coefficient increases with the increase of particle size. The frictional coefficient of block MoO_3_ was the highest, and the average value was approximately 0.129, which was close to the value of 0.126 of the base oil. Another phenomenon was found when oil with medium-sized MoO_3_ additives was supplied during the test; the friction coefficient suddenly decreased from 0.12 to 0.10 at the moment at which oil was added, and then it gradually returned to 0.12 after approximately 12 min, as shown in Figure 16b. 

The influence of particles size on friction can be explained schematically through Figure 17. As shown in Figure 17b, the medium-sized lamellar MoO_3_ is stacked by a large number of lamellar micro-MoO_3_. At the moment where oil is being added, the large number of micro-MoO_3_ particles enter the contact area, form an easy interlayer shearing sliding system and preventing direct contact in the contact area. The friction-reducing effect is even better than that of small-sized MoO_3_. Therefore, the friction coefficient is clearly reduced, as shown in Figure 16b. However, as the additives are gradually crushed during the sliding movement, the interlayer shearing sliding is reduced, and the friction coefficient returns to the original value. Compared with the large-sized MoO_3_, small-sized micro-MoO_3_ could form a more stable suspension with the base oil, and thus has an excellent friction-reducing effect, as shown in Figure 17a. Due to the large size, block MoO_3_ is easily precipitated and unevenly distributed in the base oil. Meanwhile, it is difficult for block MoO_3_ to enter the contact area of the friction pairs; thus, it does not have a good friction-reducing effect, as shown in Figure 17c.

The above results indicate that the layered structures do play an important role in the friction-reducing effect of additives. The smaller the size of MoO_3_ is, the better the friction-reducing effect that the lamellar MoO_3_ shows.

## 4. Conclusions

In order to investigate the friction-reducing ability and antiwear properties of MoO_3_ as a lubricant additive and the related lubrication mechanisms, friction and wear tests were carried out under different concentrations, particle sizes, normal loads, velocities, and temperatures, using a tribometer. The coefficient of friction, wear scar depth and width, elemental analysis of the tribofilms were discussed. The lubricant mechanism was analyzed. The conclusions are as follows.

The 0.5 wt % concentration of MoO_3_ yields the best antifriction performance, and the maximum friction reduction is 19.8% compared with that of the base oil. The 0.1 wt % concentration of MoO_3_ showed an optimal antiwear effect, and the wear scar depth is decreased by 55.9% compared with that of the base oil. The friction and wear reduction effect is comparable with those of nano-ZnO, nano-ZrO_2_, nano-CuO and nano-TiO_2_ additives [9,10,11].

The friction coefficients of the base oil and base oil with MoO_3_ additive all decreased with the increase of velocity. The friction coefficient of the base oil and base oil with MoO_3_ additives decreased and increased slightly, respectively, with the increase of normal load. 

With the addition of MoO_3_ additive, friction can be decreased considerably under a high velocity and normal load. The critical temperature of oil can be increased from 100 °C to 150 °C. 

The excellent tribological properties of MoO_3_ additives are due to the layered structure, filling of the concave areas, and the formation of the physical film and tribofilm. This research proved to be an alternative additive material in lubrication. The findings will be beneficial in its further industrial application as an oil additive in the future.

## Figures and Tables

**Figure 1 materials-11-02427-f001:**
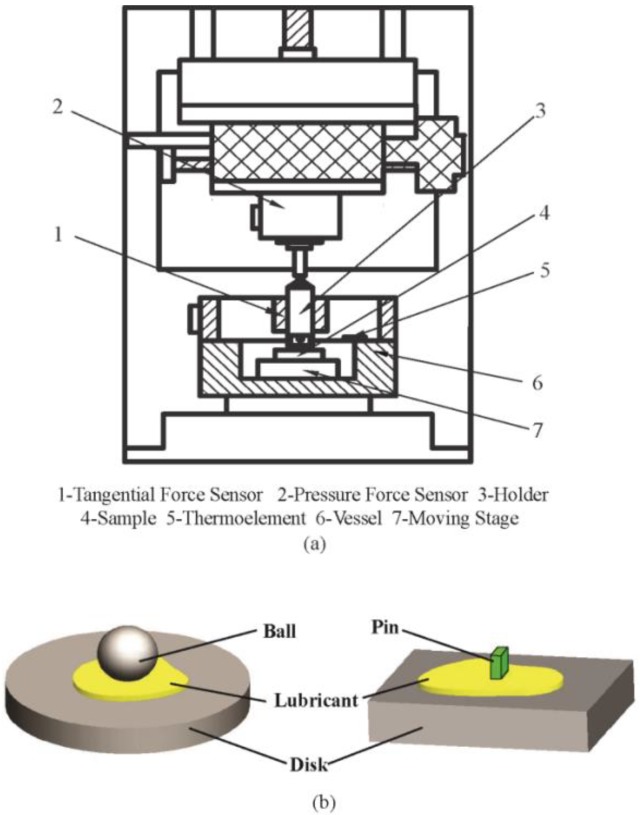
Schematic of the (**a**) testing system and (**b**) pin-on-disk and ball-on-disk configuration.

**Figure 2 materials-11-02427-f002:**
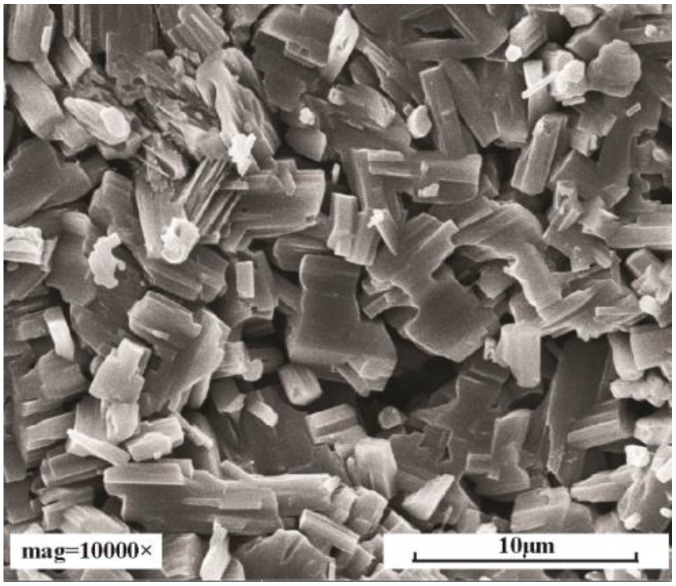
Scanning electron microscopy (SEM) image of lamellar MoO_3_.

**Figure 3 materials-11-02427-f003:**
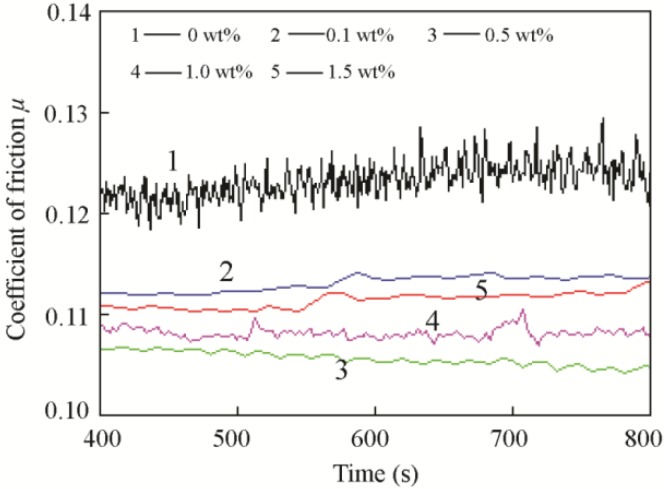
Typical friction coefficients curves of the base oil with different concentrations of MoO_3_ additives.

**Figure 4 materials-11-02427-f004:**
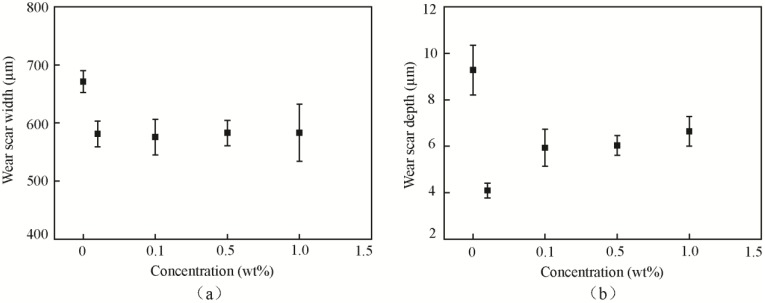
Wear scar (**a**) width and (**b**) depth of surfaces lubricated by base oil with different concentrations of micro-lamellar MoO_3_ additives.

**Figure 5 materials-11-02427-f005:**
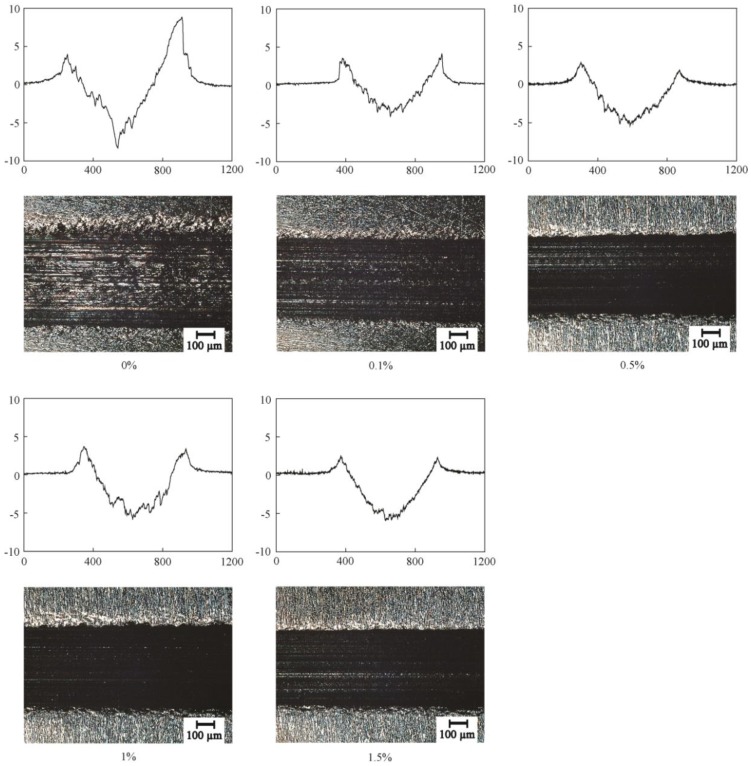
2D profile curves and high-power microscopy images of the wear scars lubricated by base oil with different concentrations of micro-lamellar MoO_3_ additives.

**Figure 6 materials-11-02427-f006:**
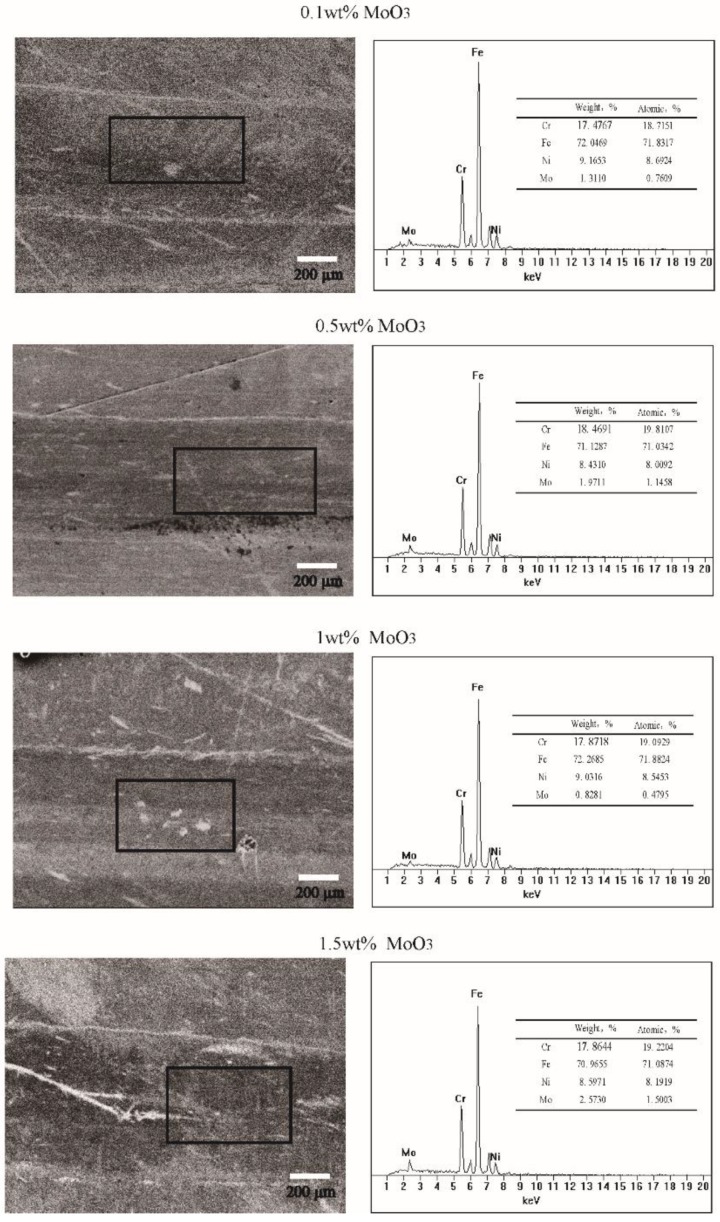
Energy dispersive X-ray spectroscopy (EDS) data of the wear scars lubricated by base oil with different concentrations of micro-lamellar MoO_3_ additives.

**Figure 7 materials-11-02427-f007:**
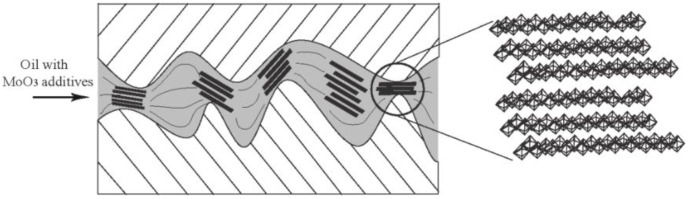
Schematic diagram of friction-reducing of micro-lamellar MoO_3_ additives.

**Figure 8 materials-11-02427-f008:**
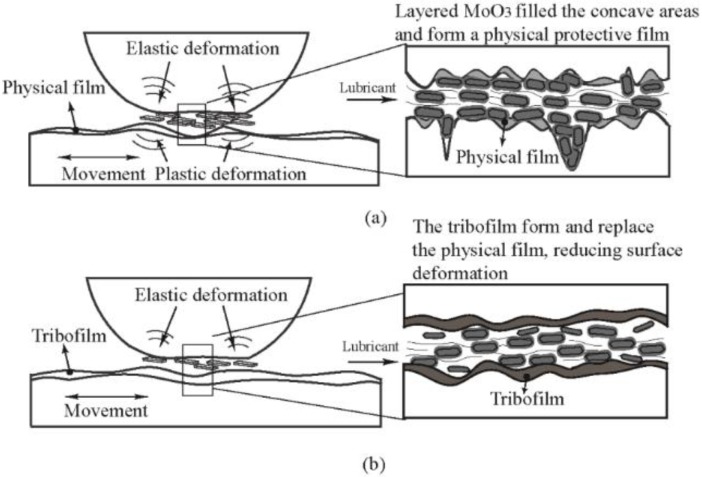
Schematic diagrams of the formations of (**a**) the physical film and (**b**) the tribofilm.

**Figure 9 materials-11-02427-f009:**
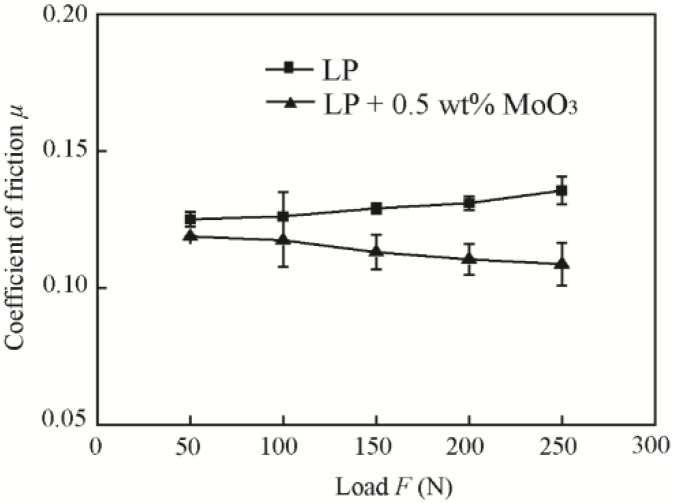
Friction coefficients of base oil and with 0.5 wt % micro-lamellar MoO_3_ additives under different loads.

**Figure 10 materials-11-02427-f010:**
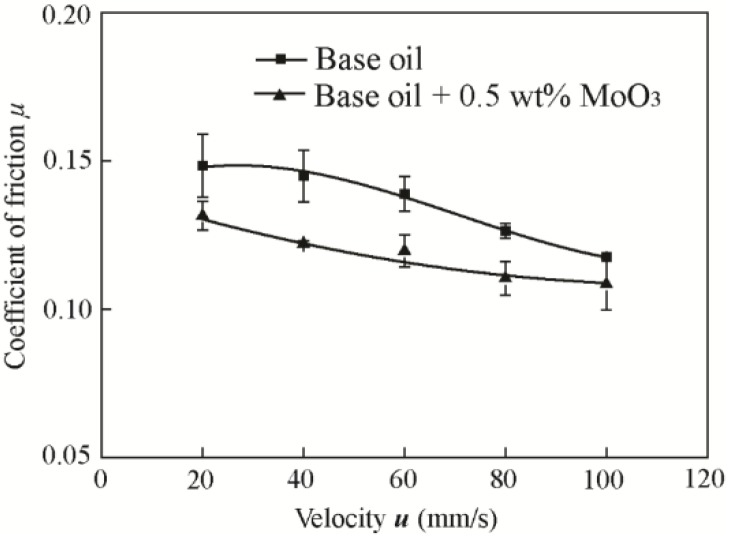
Friction coefficients of base oil and with 0.5 wt % micro-lamellar MoO_3_ additives under different velocities.

**Figure 11 materials-11-02427-f011:**
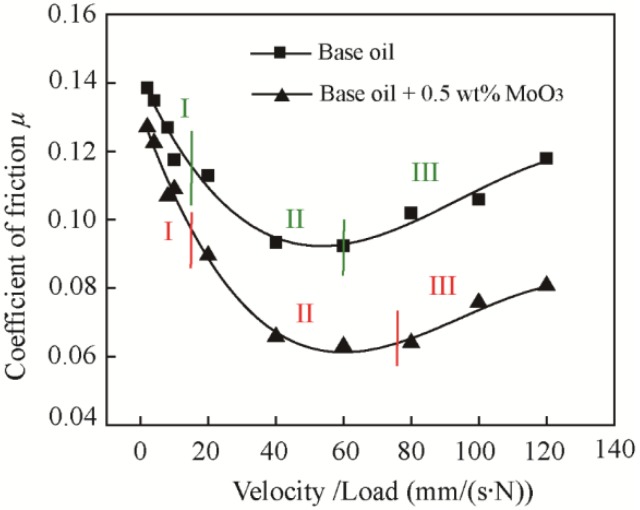
Stribeck curves of base oil and with the addition of 0.5 wt % MoO_3_. Region I: boundary lubrication; region II: mixed lubrication; and region III: hydrodynamic lubrication.

**Figure 12 materials-11-02427-f012:**
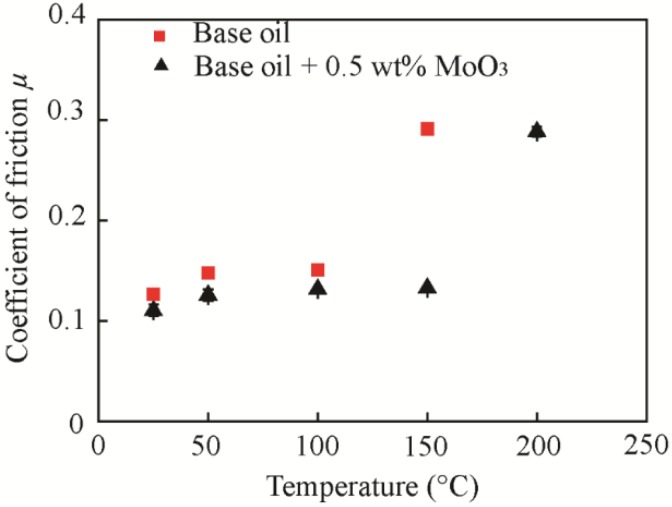
Average friction coefficient of the base oil and with 0.5 wt % micro-lamellar MoO_3_ additives under different temperatures.

**Figure 13 materials-11-02427-f013:**
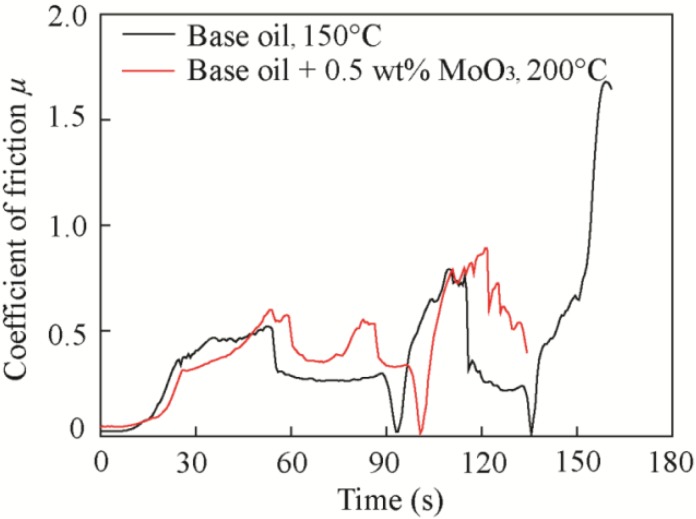
Typical friction coefficient curves at the critical temperature.

**Figure 14 materials-11-02427-f014:**
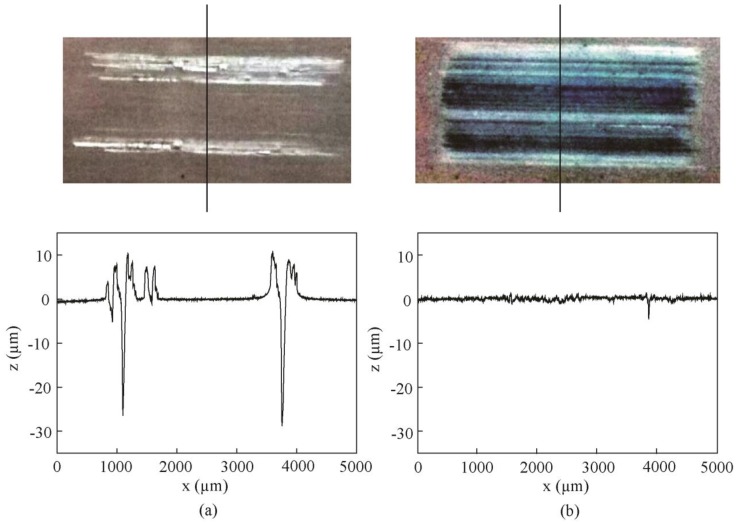
Wear scar images and 2D profile curves of samples lubricated with (**a**) base oil and (**b**) base oil with 0.5 wt % MoO_3_ at 150 °C.

**Figure 15 materials-11-02427-f015:**
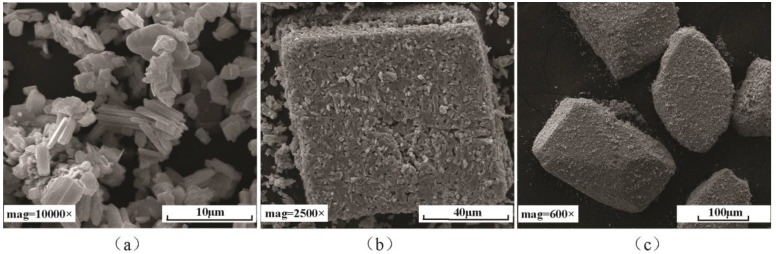
SEM images of lamellar MoO_3_ with a (**a**) small size of 2~10 μm and (**b**) a medium size of 80~100 μm, and (**c**) block MoO_3_ with a large size of 100~300 μm.

**Figure 16 materials-11-02427-f016:**
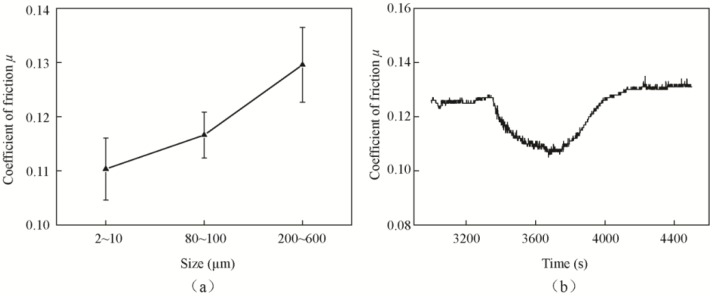
Friction coefficients of base oil with 0.5 wt % MoO_3_ additives of different sizes: (**a**) average friction coefficients and (**b**) typical friction coefficient curve of medium-sized MoO_3_.

**Figure 17 materials-11-02427-f017:**
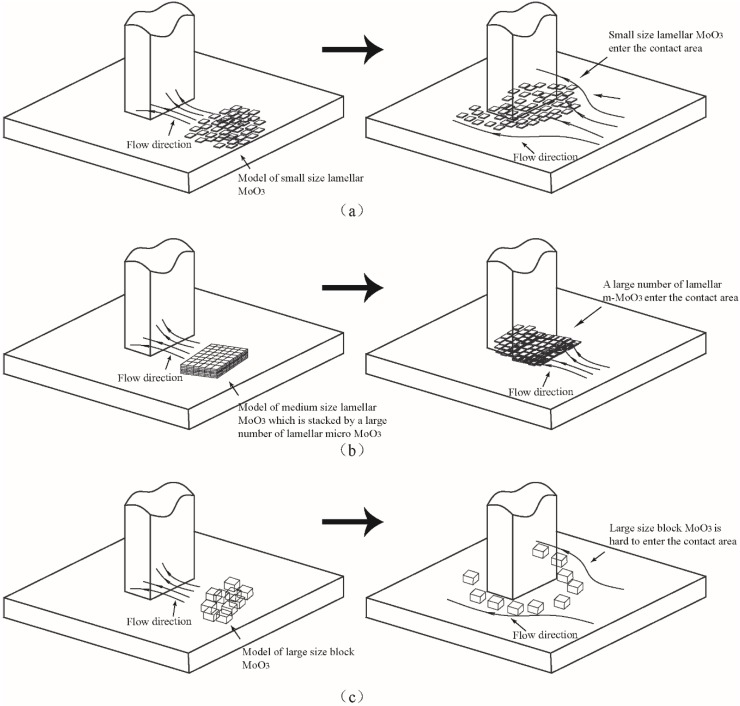
Schematic diagrams of the influence of different sizes of MoO_3_ on friction: (**a**) small-sized lamellar MoO_3_, (**b**) medium-sized lamellar MoO_3_, and (**c**) large-sized block MoO_3_.

**Table 1 materials-11-02427-t001:** Parameters of the pin-on-disk and ball-on-disk configurations.

		Material	Tensile Strength (MPa)	Yield Strength (MPa)	Hardness	Size (mm) Length × Width × Height	Arithmetical Mean Roughness *R*_a_ (μm)
Pin-on-disk	Pin	GCr15	861	518	30 HRC	5 × 2 × 5	0.06
	disk	GCr15	861	518	30 HRC	43 × 32 × 7	0.034
Ball-on-disk	Ball	9Cr18	741	295	58 HRC	*ϕ* 10	-
	disk	1Cr18Ni9Ti	680	265	158 HB	*ϕ* 40 × 10	0.034

**Table 2 materials-11-02427-t002:** Material chemical composition of the pin-on-disk and ball-on-disk configurations (weight %).

	C	Si	Mn	Cr	Ni	S	P
GCr15	1.01	0.25	0.35	1.46	-	0.002	0.012
9Cr18	1.03	0.39	0.47	18.3	0.16	0.003	0.008
1Cr18Ni9Ti	0.04	0.12	1.22	17.65	8.24	0.01	0.009

**Table 3 materials-11-02427-t003:** Average friction coefficients of the base oil with different concentrations of MoO_3_ additives.

Concentrations (wt %)	0	0.1	0.5	1.0	1.5	*p*
Coefficient of friction	0.127 ± 0.003 ^a^	0.119 ± 0.004 ^b^	0.108 ± 0.005 ^c^	0.116 ± 0.009 ^d^	0.118 ± 0.008 ^d^	0.0026

* Means and standard deviations were calculated from five repeat tests. Means in a row sharing different letters are significantly different (*p* < 0.05), and those sharing the same letter are not significantly different for each hair group (*p* > 0.05).

**Table 4 materials-11-02427-t004:** Variance analysis for friction coefficients of the base oil with different concentrations of MoO_3_ additives.

Concentrations (wt %)	0/0.1	0.1/0.5	0.5/1.0	1.0/1.5
*p*	0.0115	0.0174	0.034	0.825

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
