# Peer review of "Tribological Behavior of Lamellar Molybdenum Trioxide as a Lubricant Additive"

_materials, 2018, doi:10.3390/ma11122427_

Round 1
Reviewer 1 Report
Manuscript No.: Materials-384465
Date received: october 22, 2018
Title: Tribological Behavior of Lamellar Molybdenum Trioxide as a Lubricant Additive
Authors: Rui Liu, Xiangyong Lu, Shaogang Zhang Songyong Liu and Wei Tang*
A SHORT SUMMARY OF THE RATIONALE FOR RECOMMENDATION
According to the Abstract the paper presents an experimental tribological behavior of lamellar MoO3 as a lubricant additive under different working conditions. The related antiwear and antifriction mechanism was discussed. The results indicated that lamellar MoO3 has reduced the friction and wear due to the layered structure, filling of the concave areas, and the formation of tribofilm.
Comments and suggestion are listed in the next section.
DETAILED COMMENTS FOR AUTHOR(S):
The title and abstract are appropriate for the content of the text. Furthermore, the article is well constructed, there are many experiments done, and analyses are good enough performed.
The novel contribution is the idea of using the lamellar MoO3 as a lubricant additive to improve the tribological behavior of the oil.
Introduction.
In the introduction are presented the main elements additive to reduce friction and wear. It is mentioned that his particles “oil, slightly reduces the coefficient of friction, reduces variability and stabilizes frictional behavior“ or for instance “and nano-CuO can reduce friction and wear when used as”, “between 1 and 100 nm have good lubricating effects” and so one …. without making a clear quantitative assessment of these effects. I recommend that you give concrete values of the coefficients of friction, wear and other tribological aspects that you have generally referred to.
Experimental details
Friction and wear tests
In line 65 table 1 you intruded the parameter surface roughness Ra (μm), please give some details about this surface parameters. Why, for example, you stopped to surface characterization by Ra and not for example by RMS.
In line 73 you talked about different concentrations of MoO3 (0wt%, 0.1 wt%, 0.5 wt%, 1 wt%, and 1.5 wt%), please give some details and explanation about these percent.
In line 81 please detailed the RH 50-60% - meaning relative humidity.
Please give information about the friction coupling from the tribometer pin on disc and ball on disc, more exactly which is the material and characteristics of pin and ball (please clearly state the materials).
In line 83 and 89 you introduce the “surface roughness meter” – please describe this device maybe it’s about surface roughness tester.
In figure 3b) you present the average friction coefficients depends on concentration of MoO3 and show that the friction coefficient would have a decrease after a certain curve, which means that if I think to say a concentration of 0.75 wt% would mean from the graph that friction coefficient would be for example 0.1125, which I do not think it is in that sense.
I suggest you leave on the graph only the points that highlight the friction coefficient values you have obtained experimentally for the concentrations investigated by MoO3. Also on the graph you expressed the variability of friction coefficient based on standard deviations. Please let me how you decided to use the standard deviation for your experimental results interpretation.
Figure 4 for instance: I would appreciate very much to read in the figure caption all information needed to interpret the many details of the reported figure. Generally, the figures need to be improved are not clear enough (poor quality).
You presented in figure 5 the wear results in lubricated condition based on oil with different concentrations of micro-lamellar MoO3 additives. More qualitative and quantitative interpretations are required on the phenomena highlighted in this figure.
In line 166 you said “Second, it is known that the surface of this material is uneven and very rough on the micro-and nanoscales” – what material is involved???
In Figure 8 is represented the schematic diagrams of the formations of (a) physical film and (b) tribofilm. My question is about the surface that performs the movement from (b), it should not have exactly the same shape as the one in the figure (a) (to have the same amplitude and frequency of the irregularities on the surface to ensure that this surface is fit with the upper one)
Regarding the Influence of the normal load and velocity in this section 3.2 lines 189 you take into consideration only the base oil with 0.5 wt% micro-lamellar MoO3 additives under different loads and speeds. Why no attempts were made for all samples ie for all types of MoO3 oil additive presented as work object at the beginning of the paper?
In line 221 you said that “Ra1 and Ra2 are the surface roughnesses of the contact surfaces”, I think it would be more correct is roughness parameter - arithmetical mean height that indicated the average of the absolute value along the sampling length.
If in lines 227 – 230 you introduce in relations (3) and (4) the relative velocity of the two friction pairs denote u, and F is the load, it is necessary to use the same symbolization and put them on the x-axis in the graphs shown in the figures 9 and 10.
Also please indicate the mathematical expression for E* - the equivalent elastic modulus.
Please give additional explanations about this “Moreover, the layered lubricant additive can reduce the viscosity of the oil significantly with the increase of shear rate [9], which can also help to reduce the inner friction of the oil film and improve lubrication”, more exactly how to reduce the viscosity of the oil significantly with the increase of shear rate?
In line 243 if you agree please replace: “50 r/min to 300 r/min” with 50 rpm to 300 rpm or 50 rot/min to 300 rot/min.
In figure 11 you present the Stribeck curves of base oil with the addition of 0.5wt% MoO3 and in you interpreted the friction coefficient against the speed/force, but the maximum value considered for the friction coefficient is much higher than the one obtained so far which was below 0.15 (see figure 9 and 10).
For section 3.4 Influence of the temperature, please describe how did you monitor the test temperature? And for the chart in Figure 12 makes the same observation that I did before we cannot talk about joining the points that indicate the coefficient of friction at a certain temperature through a curve.
More elaborate conclusions should be given, including comparisons with the literature.
None of methods can be considered original, nor are the motivations and goals of the experimental efforts very well provided to the reader. Overall, it is an important work, and should be considered of interest for the audience of Materials.
In closing, again, I appreciate the opportunity to have read your work. I look forward to seeing more of it in the future. Best of luck!
Author Response
We would like to thank the reviewer for a thorough review and useful comments. All changes suggested by the reviewer have been incorporated. We have spent considerable time in making changes and we hope that the referee would find them to be satisfactory. If there are further changes to be made, we will be happy to comply.
The point-by-point response to the reviewer’s comments has been update in the attached file.

Reviewer 2 Report
Review
Manuscript ID: materials-384465
In this study, the tribological properties of micro-lamellar MoO3 as a lubricant additive were investigated.
The manuscript is not well written and organized. The introduction must be reviewed. The experimental methodology used is not explained appropriately. The results are not clear.
The abstract is expected to include a brief digest of the research, that is, new methods, results, concepts, and conclusions only. The abstract needs to be more focused. The experimental methodology and achievements needs mentioned clearly. The weaknesses of the manuscript are listed below
The references in the text are not in accordance with the journal’s guidelines. Please check it at: https://www.mdpi.com/journal/materials/instructions.
Introduction must provide a comprehensive critical review of recent developments in a specific area or theme. Introduction is expected to have an extensive literature review followed by an in-depth and critical analysis of the state of the art. Bibliographic references must be explained individually and not grouped. In this section it would be opportune to introduce the experimental methodology that will be used in the manuscript with respect to the mentioned bibliographic references. Describe how the results will be presented. I suggest add information to better describe what other researchers have done in this area. I recommend rewriting the introduction.
When the authors talk about the tribometer, I don't understand if it works with Reciprocating or continuous motion? Please better specify this one in the abstract, introduction and in the section 2.
It would be appropriate in section 2.1 only to indicate that the tests were carried out under lubricated conditions and move the sentence of line 72-73 and lines 77-78 in section 2.1.
In section 2.1 I do not understand if the authors have carried out the tests by combining the different parameters (Loads, speeds and temperatures) or if for each load a different speed and temperature have been chosen. Explain it better in the text, perhaps graphically (table or flow chart). Furthermore, why are these parameters (Loads, speed and temperature) chosen for the tests? Explain it in the text and introduce references if appropriate. The same occurs with the chosen parameter for the wear test. Explain it in the text with the use of any bibliographic references.
In the section 2.2 What are the parameters used for the characterization of the surfaces of the specimens? I refer to magnification, for example, of the microscope. And those of spectroscopy? Here too it would be appropriate to introduce this information and if necessary add necessary references.
In section 2.3, the authors do not explain why they use these concentrations for the lubricant. They do not add any feedback with previous studies.
The results section is chaotic. I also do not understand why the authors use different test conditions between the Friction test and the wear test. Any previous work that justifies this choice? Please mention it in the text.
The quality of the graphs in the results section is very bad. The curves are illegible. They are also grainy. The quality of the figures should be improved.
The manuscript does not say how many tests were performed for each load condition, speed and temperature and for each concentration. I hope it's not one for each test. In this case, there would be problems linked to the repeatability of the test. How many tests have you performed for each dynamic condition?
In section 3.3 it is not clear how the authors plotted the Stribeck curve. In fact, if it is a reciprocating tribometer the curve of Stribeck should be plotted as explained in:
A Ruggiero, R D’Amato, M Merola, P Valašek, M Müller, Tribological characterization of vegetal lubricants: Comparative experimental investigation on Jatropha curcas L. oil, Rapeseed Methyl Ester oil, Hydrotreated Rapeseed oil. Tribology International 109 (2017), 529-540.
The conclusions highlighting the reasons for the research, summarizing the steps followed for the development of the results and highlighting the results obtained. Rewrite the conclusions.
For this reasons i suggest to reject this manuscript for the publication.
Author Response

(The authors gave the same response as above.)

Reviewer 3 Report
Some corrections should be done in the submission, namely:
Section 2
1) what does it mean “43 × 32 × 7” for a disk in Table 1?
2) a diameter should be marked as Æ.
3) where are a height, a width and a length in 5 × 5 × 2 pin?
4) what method was used to achieved Ra 0.034? Such Ra values are untypical for friction surfaces
5) the pin (s?), ball and disk should be shown in Fig. 1; the actual figure is uninformed
6) the method used to mix clean oil and MoO3 particles should be described in this section
8) how different oil temperatures were controlled and maintained?
9) the wear measuring method using a surface roughness meter should be described
10) why the base oil but not machine oil was used?
Section 3
11) research conditions were already described in section 2 and should not be described in section 3 once more
12) how measuring of wear scar sizes with a tolerance of 0.01 µm was secured (please, see Figure 4)?
13) curves instead of columns should be used in Figure 4, because tested factors are continuous variables
14) to assert that plastic deformations occur on the contact surface, significantly larger increases than in Figure 4 should be used
15) all images in figures should be mo contrast
16) 217 line: “The ratio of film thickness to surface roughness λ is usually used to determine the types of lubrication” – please, add the reference
17) what method was used to calculate regression equations?
18) most references are relatively old, please, add more sources of 2016-2018 years.
Author Response

(The authors gave the same response as above.)

Round 2
Reviewer 2 Report
The topic of this paper is relevant, and of interest to the audience of this journal. The research is based on rigorous academic standards, and the state fo the art, methodology and results have been improved after reviewing. The paper has been improved and the authors have corrected the mistakes discussed. The paper is ready to be published.